# Insights on the Functions and Ecophysiological Relevance of the Diverse Carbonic Anhydrases in Microalgae

**DOI:** 10.3390/ijms21082922

**Published:** 2020-04-22

**Authors:** Erik L. Jensen, Stephen C. Maberly, Brigitte Gontero

**Affiliations:** 1Aix Marseille Univ, CNRS, BIP, UMR 7281, IMM, FR3479, 31 Chemin J. Aiguier, CEDEX 20, 13 402 Marseille, France; erikjensenrojas@gmail.com; 2UK Centre for Ecology & Hydrology, Lake Ecosystems Group, Lancaster Environment Centre, Library Avenue, Bailrigg, Lancaster LA1 4AP, UK; scm@ceh.ac.uk

**Keywords:** carbonic anhydrase, microalgae, carbon dioxide, CO_2_-concentrating mechanisms

## Abstract

Carbonic anhydrases (CAs) exist in all kingdoms of life. They are metalloenzymes, often containing zinc, that catalyze the interconversion of bicarbonate and carbon dioxide—a ubiquitous reaction involved in a variety of cellular processes. So far, eight classes of apparently evolutionary unrelated CAs that are present in a large diversity of living organisms have been described. In this review, we focus on the diversity of CAs and their roles in photosynthetic microalgae. We describe their essential role in carbon dioxide-concentrating mechanisms and photosynthesis, their regulation, as well as their less studied roles in non-photosynthetic processes. We also discuss the presence in some microalgae, especially diatoms, of cambialistic CAs (i.e., CAs that can replace Zn by Co, Cd, or Fe) and, more recently, a CA that uses Mn as a metal cofactor, with potential ecological relevance in aquatic environments where trace metal concentrations are low. There has been a recent explosion of knowledge about this well-known enzyme with exciting future opportunities to answer outstanding questions using a range of different approaches.

## 1. Introduction

Microalgae are a heterogenous group of eukaryotic and prokaryotic microorganisms that are almost exclusively photosynthetic, phylogenetically distinct, and evolved at different geological periods during the Earth’s history [1]. The earliest prokaryotic photosynthetic microalgae, cyanobacteria, appeared more than 2.3 billion years ago [2]. Eukaryotic microalgae, including red and green lineages, evolved over 1.5 billion years ago as the result of an endosymbiosis event between a photosynthetic cyanobacterium and a heterotrophic eukaryote [3]. Other microalgae are the result of additional endosymbiotic events between photosynthetic microalgae and photosynthetic or non-photosynthetic hosts. Therefore, some microalgae are the result of a secondary (e.g., the stramenopiles), tertiary, or even quaternary (e.g., some dinoflagellates) endosymbiosis [4,5]. Microalgae are important primary producers in aquatic environments. Their success is in part the result of highly efficient photosynthetic CO_2_ fixation, which can be 10 to 50 times higher than that of land plants [6]. Moreover, this high efficiency can occur even in environments where CO_2_ is usually limiting, because microalgae have evolved different strategies to take up and fix CO_2_, all of which involve the enzyme carbonic anhydrase (CA; EC 4.2.1.1). Changing concentrations of CO_2_ and O_2_ on Earth have altered the availability of inorganic carbon for photosynthesis and influenced the evolution of microalgae [7].

CAs catalyze the reversible reaction of CO_2_ hydration to form HCO_3_^-^, which is otherwise slow, and they occur in all kingdoms of life. They have several functions including maintenance of the acid–base balance, lipid biosynthesis, and the uptake of inorganic carbon [8]. So far, eight different CAs classes have been described (α-, β-, γ-, δ-, ζ-, η-, θ-, including a recently described ι-CA; Table 1) [9,10,11,12,13,14]. Although all classes of CA catalyze the same reaction, they seem to be unrelated, since they share little or no sequence or structural similarity. All CAs are metalloenzymes that commonly use Zn^2+^ as a metal ion cofactor; however, it is known that some classes are cambialistic and/or replace Zn^2+^ with other metals, such as Cd^2+^, Co^2+^, Fe^2+^, or Mn^2+^ [15]. Seven of the eight classes of CA (currently excluding η-CA found within *Plasmodium* sp [16]) occur within microalgae, potentially reflecting their evolutionary diversity but also their reliance on this enzyme for carbon fixation. In fact, some CAs classes were first discovered in microalgae (i.e., diatoms) [11,12,17]. Moreover, the class and subcellular distribution of CAs within microalgae cells can also vary among species, even in those belonging to the same family (Figure 1).

In this review, we focus on the importance of CAs for inorganic carbon uptake in microalgae, on their role in the algal CO_2_-concentrating mechanisms, on their diversity, and how these enzymes help microalgae to survive and flourish in modern aquatic environments.

## 2. The Need for CO_2_-Concentrating Mechanisms in Microalgae

Aquatic environments are commonly limited by CO_2_ and so aquatic organisms cannot rely solely on passive CO_2_ uptake. For instance, in the oceans where the pH is between 7.8 and 8.4, only approximately 1% of the total dissolved inorganic carbon (DIC) is present in the form of CO_2_, bicarbonate being the most abundant inorganic carbon compound [25,26,27]. Moreover, at equilibrium with the atmosphere, the concentration of CO_2_ is around 15 µM, which is lower than the typical values of half saturation constant (^C^*K_0.5_*) of the CO_2_-fixing enzyme ribulose 1,5-bisphosphate carboxylase oxygenase (Rubisco; ^C^*K_0.5_* = 100–180 µM in cyanobacteria [28], 23–65 µM in diatoms [29], and 15–24 µM in haptophytes [30]). In addition, Rubisco-mediated carboxylation competes with the oxygenation of ribulose 1,5-bisphosphate (RuBP), which reduces carbon fixation and promotes photorespiration [31]. The extent to which these two competitive reactions occur depends on the O_2_ and CO_2_ concentrations at the active site of Rubisco and the relative affinity of the enzyme for the two gases. Thus, a CO_2_ uptake strategy based only on the diffusion of CO_2_ from the extracellular milieu to the chloroplast will restrict CO_2_ fixation rates at atmospheric CO_2_ concentrations. In fresh waters, CO_2_ concentrations substantially above and below air equilibrium can occur [32], providing an opportunity at high concentrations for species with relatively low abilities to exploit inorganic carbon [33] but producing more challenging inorganic carbon conditions when the concentration of CO_2_ is very low.

To cope with CO_2_ limitation, photosynthetic organisms evolved CO_2_-concentrating mechanisms (CCMs) to increase the concentration of CO_2_ in the vicinity of the Rubisco active site [34,35]. CCMs have been studied in several photosynthetic microalgae, including green algae [36,37,38], cyanobacteria [39], and diatoms [34,40]. A biochemical CCM involves the pre-fixation of CO_2_ into C_4_ organic compounds (i.e., oxaloacetate), which is a process termed C4 photosynthesis that occurs in 3% of terrestrial higher plants (e.g., *Zea mays*). This C4 metabolism is present in some aquatic plants (e.g., *Hydrilla verticillata* and *Ottelia alismoides* [41,42,43]) and in some green macroalgae (e.g., *Ulva*
*prolifera* [44]). In contrast, although C4 photosynthesis has been proposed in the diatom *Thalassiosira weissflogii* (now called *Conticribra weissflogii*), more recent data indicate that this type of CCM is not widespread in microalgae [38,45,46,47,48,49,50].

Biophysical CCMs involving the active transport of inorganic carbon into the cell as HCO_3_^−^ or CO_2_ are more frequent than biochemical CCMs in microalgae [51,52]. Cyanobacterial CCMs are based on the transport of CO_2_ or HCO_3_^−^ and are extremely efficient, as they can elevate the CO_2_ concentration around Rubisco embedded within a polyhedral compartment called a carboxysome, 1000 times above the external CO_2_ concentration [39]. CCM components have been described both in marine and freshwater cyanobacteria and recently in cyanobacteria living in alkaline lakes [53]. Most bicarbonate transporters belong to the solute carrier family (SLC) and have been well described in mammals and humans [54]; however, other types of transporters have also been found in photosynthetic organisms. The first HCO_3_^−^ transporter described in cyanobacteria was BCT1 from *Synechococcus* [55]. It belongs to the ATP-binding cassette (ABC) transporter family and has a K_0.5_ for HCO_3_^−^ of 15 µM [55]. It is encoded in the cmpABCD operon, which is highly induced by low CO_2_ [56]. However, all strong alkaliphilic cyanobacteria lack this transporter [53]. Other transporters described in cyanobacteria include the two Na^+^-dependent HCO_3_^−^ plasma membrane transporters SbtA (a sodium-dependent HCO_3_^−^ symporter; with a K_0.5_ for bicarbonate < 5 µM and low flux of HCO_3_^−^ uptake), which was shown recently to be allosterically regulated [57], and BicA (a sulfate permease or SulP-type sodium-dependent HCO_3_- transporter) that has a low affinity for HCO_3_^−^ (K_0.5_ =70–150 µM) and a high flux of HCO_3_^−^ uptake [55,58]. Another cyanobacterial system comprises two thylakoid CO_2_ uptake systems based on NAD(P)H dehydrogenase type 1 (NDH-1_3_/_4_) that are also induced by carbon limitation [59,60,61].

Eukaryotic microalgal bicarbonate transport is more complex than that of cyanobacteria, since the eukaryotic cell contains organelles surrounded by membranes. In the green alga *Chlamydomonas reinhardtii*, two plasma membrane transporters, high-light activated protein, HLA3 and low CO_2_ inducible protein, LCI1, have been well studied; more recently, a new transporter, CIA8 (for Ci accumulation), was shown to be needed for optimal growth at low CO_2_ and for bicarbonate uptake and could be located within the thylakoid membrane [62]. Chloroplast membrane transporters include the LCIA (also known as NAR1.2), which is known to form a complex with the soluble protein LCIB [63]. Two proteins, CCP1 and CCP2, with similarity to mitochondrial carrier proteins are believed to be involved in bicarbonate uptake in the chloroplast. However, the mutants of these proteins do not show an abnormal carbon concentration phenotype; thus, their involvement in CCM is not yet clear [64].

In diatoms, bicarbonate uptake has been studied in the two model species *Phaeodactylum tricornutum* and *Thalassiosira pseudonana*. The inorganic flux of carbon in diatoms has been estimated to increase CO_2_ around Rubisco up to 60 µM, just above the half-saturation concentration [35]. A plasma membrane-bound bicarbonate transporter from *P. tricornutum* belongs to the solute carrier 4 family (SLC4) and seems to be closely related to a human type rather than those found in green algae [65]. Moreover, three out of seven SLC4 genes found in *P. tricornutum* are induced by low CO_2_ and are highly inhibited by the anion exchange inhibitor 4,4’-Diisothiocyanostilbene-2,2’-disulfonic acid (DIDS) [65]. SLC4 family proteins are also believed to occur in *T. pseudonana* as gene homologs to those in *P. tricornutum* have been found [66]; however, their function has not been studied yet.

## 3. Diversity of Carbonic Anhydrases in Microalgae and Their Importance in the CCM

Carbonic anhydrases are widely distributed in photosynthetic organisms [10] and play an important role in HCO_3_^-^/CO_2_ uptake and in the concentration of CO_2_ around Rubisco in microalgae. α-CA was first discovered in erythrocytes and widely studied in mammals [67,68,69,70], but it is also present in higher plants, algae, and cyanobacteria, among other organisms within all the kingdoms of life. α-CAs are often more active than other classes of CA [10] and are typically found as monomers; however, one type, in the fungus *Aspergillus oryzae,* is a dimer [71] and two α-CAs in *C. reinhardtii* (CAH1 and CAH2) are tetramers [72]. In *C. reinhardtii,* 12 genes that encode CA isoforms, including three alpha, six beta, and three gamma or gamma-like CAs [73], have been reported and more recently, three more beta CAs have been described [74] (Figure 1). Another extracellular α-CA, EcaA, from cyanobacteria is also found in the periplasmic space; however, it does not have a role in the CCM [75,76]. In the marine eustigmatophyte *Nannochloropsis oceanica*, the CAH1 located in the lumen of the epiplastid endoplasmic reticulum is also essential for the CCM [77]. Other α-CAs having a possible role in CCMs are found in many other photosynthetic organisms, including haptophytes, rhodophytes, phaeophytes, and cryptomonads [10].

β-CA was discovered in chloroplasts from leaves [78]. They are present in most photosynthetic organisms, and in some non-photosynthetic microorganisms, but not in animals [74]. In contrast to the α-CAs, β-CAs are more frequently found in different oligomeric states, and several trimeric and tetrameric structures have been reported [79,80,81]. This class of CA is the best studied in land plants, and the role of β-CA in the CCM of terrestrial C4 plants has been well established, where it is mainly expressed in the cytoplasm of mesophyll cells, thus providing the HCO_3_^−^ required for phosphoenolpyruvate carboxylase [18,82]. β-CA is highly expressed in the green tissues of C3 plants, suggesting a possible role in photosynthesis [83], and in *Arabidopsis thaliana*, two cytoplasmic β-CAs are essential for growth at low CO_2_ [84]. In cyanobacteria containing α-carboxysomes, a β-CA (CsoSCA) is present in the carboxysome shell [85] (Figure 1); this CA was first proposed as a new sub-class, which was called ε (epsilon), but later it was shown that the ε-CA is just a modified β-CA, and this denomination is no longer used [86,87].

The γ-CA is a trimeric protein first described in Archaea [88], but it is also present in photosynthetic organisms [10]. In the cyanobacterium *Synechococcus* PCC7002, the protein CcmM, a component of the β-carboxysomes, is a γ-CA that shares 34%–36% identity with the active site of that from the anaerobic methane-producing species *Methanosarcina thermophila* from the Archaea domain with the amino acids involved in Zn^2+^ coordination perfectly conserved [75]. Although the Archaean type is able to replace Zn^2+^ by Fe^2+^ [89], this feature has not been observed in photosynthetic organisms. In contrast, in microalgae and plants, γ-CAs are located in mitochondria, and it is not clear whether they are involved in the CCM or not [18]. However, Wei et al. [90] reported that the RNAi-based down-regulation of a γ-CA in *N. oceanica* significantly decreased cell growth in cells grown at air CO_2_ and affected photosynthetic oxygen evolution; thus, this CA is likely to play a role in the CCM, although it is predicted as a mitochondrial protein. So far, the CA activity of other putative γ-CAs from eukaryotic algae has not been shown, and more work is needed to determine their physiological role and possible involvement in CCMs.

Three of the eight known CA classes were first discovered in diatoms [11,12,17,91]. In addition, the highly variable number, classes, and distribution of CAs in diatoms is unique compared to other organisms and might be related to their complex evolutionary origin [92,93]. The subcellular localization of several CAs present in diatom genomes has been predicted or experimentally demonstrated [66,94,95] (Figure 1). In *P. tricornutum,* five α-CAs are located in the four-layered chloroplast membrane, while in *T. pseudonana,* only one is present in the chloroplast stroma [95]. In addition, only two β-CAs have been found in the pyrenoid of *P. tricornutum* and none have been found in *T. pseudonana*. Two γ-CAs are also found in the mitochondria of *P. tricornutum* and three have been found in *T. pseudonana*; in the latter species, there is also a γ-CA in the cytoplasm. However, as in other algae, the activity and the role of the γ-CA in the diatom CCM has not been studied.

The δ-CA and the ζ-CA were both first discovered in the marine diatom *T. weissflogii* [17,91]. The δ-CA has been observed in algae derived from secondary endosymbiosis and in prasinophytes [10]. Surprisingly, the δ-CA is a cambialistic CA, where Zn^2+^ can be replaced by Co^2+^ at the active site [21], and in the ζ-CA, Zn^2+^ can be replaced by Cd^2+^ [96]. The CDCA (ζ-CA) from *T. weissflogii* has a catalytic efficiency (k_cat_/K_m_) of 8.7 × 10^8^ M^−1^ s^−1^ when Zn^2+^ is bound and 1.4 × 10^8^ M^−1^ s^−1^ when Cd^2+^ is bound [23]. These values are comparable to other α-CAs, such as the human CAII (k_cat_/K_m_ = 1.5 × 10^8^ M^−1^ s^−1^) [97]. The ζ-CA occurs naturally in several diatom species, including those from the genus *Thalassiosira* [98].

θ-CA is found in the lumen of the thylakoids in *P. tricornutum* [11]. It is likely that its role is to convert HCO_3_^-^ into CO_2_ inside the thylakoid lumen [10]. In addition, when the mRNA for θ-CA was silenced in *P. tricornutum*, the growth rate was lower than that of the wild type at air equilibrium and at high CO_2_ concentration, suggesting that this CA plays a fundamental role in photosynthesis and not just in the CCM [11]. The LCIB from *C. reinhardtii* was originally described as a β-CA [24] but subsequently classified as an ortholog of a θ-CA with a Cys-Gly-His rich (CGHR) domain [99]. It is found around the pyrenoid and might be involved in the prevention of CO_2_ leakage from the pyrenoid to the chloroplast stroma [24,100]. BLAST (Basic Local Alignment Search Tool) analysis shows that the θ-CA is also present in the diatoms *T. pseudonana, T. oceanica, Fragilariopsis cylindrus,* and *Fistulifera solaris*, suggesting that this CA could be widespread among diatoms (Figure 2) [1].

Finally, the most recently discovered class, ι-CA, was highly expressed in *T. pseudonana* grown at low CO_2_ [101,102]. This class uses Mn^2+^ as a cofactor, while no activity has been observed in the presence of Zn^2+^ or other metals [12]. In addition, it has been shown that ι-CA plays a role in the diatom CCM, as it dramatically increases the affinity for CO_2_ when overexpressed [12]. Moreover, it has been proposed to be located in the periphery of the chloroplast, presumably in the chloroplast endoplasmic reticulum (CER) boundary (Figure 1). Sequence comparison and phylogenetic analyses show that ι-CA is widely distributed in marine phytoplankton, including other diatom species, haptophytes, cryptomonads, and pelagophytes [1,12].

## 4. Regulation of CA Expression

In the green alga *C. reinhardtii*, the two periplasmic α-CA, CAH1 and CAH2, show an opposite regulation upon environmental CO_2_: CAH1 is more abundant under low CO_2_, while CAH2 is more abundant under high CO_2_ [72]. In *N. oceanica*, CAH1 was also more abundant in cells cultured at low CO_2_ [77]. Blanco-Rivero et al. (2012) showed that after *C. reinhardtii* cells are transfered to low CO_2_, the thylakoid luminal CAH3 becomes phosphorylated, more active, and is relocated from the photosystems II area where it is associated to its electron donor side toward the pyrenoi—a specialized compartment inside the chloroplast that is analogous to the carboxysome present in cyanobacteria, where Rubisco is packed together with carbonic anhydrases [103]. As has been observed for CAH3, it is likely that post-translational modifications triggered by low CO_2_ are also present in other α-CAs. In contrast, the two closely related β-CAs, CAH7 and CAH8 (Figure 1), are not regulated by CO_2_ [104].

The transcriptional regulation of CA expression has been studied in *C. reinhardtii*. The expression of the low CO_2_-inducible CAH3 appears to be tightly regulated by the master regulator of the CO_2_ response CCM1 (CIA5) [105]. Another gene, encoding for CAH1, is also a CO_2_-responsive gene regulated by the Myb transcription factor LCR1 [106]. However, no homolog proteins of CCM1 and LCR1 have been described in stramenopiles, alveolates, or haptophytes. In diatoms, it is believed that a transcriptional regulation of the response to CO_2_ might exist, which is mediated by the second messenger cAMP [107]. In fact, a bZIP transcription factor in *P. tricornutum* (ptbZIP11) is shown to bind specifically to a CO_2_-cAMP responsive element in the regulatory region of the *ptca1* gene, encoding for the pyrenoidal β-CA [108].

The patterns of CA expression in response CO_2_ concentration and their role in CCMs differ among species. For example, in the coccolithophore *Emiliania huxleyi*, the transcript of a δ-CA is highly expressed, but it is not affected by the presence of CO_2_, and its role in the CCM is unclear [109]. In the marine dinoflagellate *Lingulodinium polyedrum*, it is unclear whether the expression of an external δ-CA is regulated by CO_2_ concentration; however, this enzyme is important in increasing CO_2_ availability at the cell surface [110]. In contrast, the δ-CA from *T. weissflogii* and *T. pseudonana* (found in the periplasmic space) are highly up-regulated in cells grown at low CO_2_ [101,111]. In addition, in *T. pseudonana*, McGinn and Morel (2008) showed that the expression of two isoforms of the δ-CA (Tp1 and Tp2) is down-regulated at the level of protein and transcripts at low concentrations of Zn or Co. A similar pattern of regulation in response to low CO_2_ and Zn concentration was observed for the Cd-containing ζ-CA (CDCA) [22].

Since many CAs participate actively in photosynthetic CO_2_ fixation, it is not surprising that their activity is coordinated with diel changes in light availability and the activity of the whole photosynthetic machinery. Many responses to light and dark are mediated by redox conditions within the cell. Two major redox systems exist, based on thioredoxin or glutaredoxin, that are ubiquitous and able to regulate several cellular processes through dithiol–disulfide exchanges with proteins. In *P. tricornutum*, the two pyrenoid-localized β-CAs (Figure 1), named PtCA1 and PtCA2, are regulated by thioredoxins [112,113], while in *A. thaliana*, the chloroplastic β-CA (ID: At3g01500) is regulated by glutaredoxins [112,113]. In addition, PtCA1 is also regulated by CO_2_ and light [114,115], suggesting that in diatoms, CO_2_ uptake in the pyrenoid might be controlled by thioredoxins. Moreover, transcriptional regulation may also occur, since the mRNA expression of *ptca1* is down-regulated at high CO_2_ concentration at the promoter level [115,116]. The CAH8 from *C. reinhardtii* is homologous to the β-CA in *A. thaliana* and thus it might be redox regulated, but this needs to be investigated further. However, redox regulation of β-CA also occurs in non-photosynthetic organisms. For example, oxidation of the β-CA from the pathogen *Mycobacterium tuberculosis* can trigger the formation of a disulfide bond at the active site involving the cysteine residues that coordinate the catalytic metal ion, thereby leading to its inactivation [117]. In this case, oxidation can be triggered either by air and H_2_O_2_ but also by some oxidative compounds derived from the host.

## 5. Other Roles of CAs in Microalgae

Carbonic anhydrases are involved in other physiological and metabolic processes in cells, other than the CO_2_-related metabolism described above. In fact, the first function reported for a CA was on the transport and elimination of CO_2_ through blood in mammals [118,119]. CAs have been also shown to be essential in acid–base homeostasis in different organisms, including mammals, fish, and arthropods [120]. In plants, the mutation of genes encoding for β-CAs affects the CO_2_ and environmental-induced stomatal response in *A. thaliana* and *Z. mays* [121,122]. Other roles of CAs in plants include plastid lipid biosynthesis [123] and the regulation of carbon metabolism during root development [124].

Little is known about other roles of CAs in photosynthetic microalgae. A *cia1* mutant *C. reinhardii*, which is known to have a high requirement for CO_2_ and carrying two mutations in the thylakoid luminal α-CA CAH3, is much less effective at carrying out fatty acid (FA) desaturation in the thylakoid membrane. A role of this CA in the control of the composition of the photosynthetic membrane has been proposed [125]. In fact, highly unsaturated FAs are believed to be important for the normal function of photosystems I and II by affecting membrane fluidity [126]. Mackinder *et al.* (2017) showed that CAH3 also interacts with the proteins from the twin-arginine translocation (tat) pathway, TAT2 and TAT3, and so CAs may also be involved in the transport of proteins to the thylakoid lumen [127].

Chemotaxis towards HCO_3_^-^ has been demonstrated in *C. reinhardtii* [128] and toward CO_2_ for five species of microalgae [129]. In this regard, the β-CA, CAH6, is expressed in the flagella of *C. reinhardtii*, and thus, it might be involved in chemotactic movements triggered by environmental inorganic carbon concentrations [127].

A role of mitochondrial CAs in anaplerotic carbon incorporation into the tricarboxylic acid (TCA) cycle has been proposed, in which HCO_3_^-^ is supplied to the cytosolic phosphoenolpyruvate carboxylase (PEPC) of *C. reinhardtii* [130]. In contrast, diatoms have no predicted cytosolic PEPC, but a mitochondrial PEPC (PEPC2) is predicted in *T. pseudonana* and *P. tricornutum* instead [131], suggesting that the same anaplerotic role of mitochondrial CAs is also likely to occur in diatoms. This is even more relevant as these reactions occur in the same compartment; however, this needs to be investigated further.

## 6. Microalgal CAs and the Ecology of Aquatic Environments

The discovery of CAs that are capable of replacing zinc by other metal ions is of significant relevance, as it reveals an adaptation to the availability of trace metals, and this might have contributed to the ecological success of many photosynthetic microalgae, especially diatoms. This has been extensively reviewed recently by Morel et al. (2020) [15], so we focus specifically on Mn, which is the cofactor of the newly discovered ι-CA [12].

In marine environments, particularly in coastal waters and estuaries, manganese is more abundant than zinc by up to 20–50 times [132,133]. In addition, Mn^2+^ is a cofactor of several other enzymes, such as the Mn-superoxide dismutase (MnSOD) [134], and it is an important component of the photosystem II reaction center [135,136]. Indeed, in *T. pseudonana*, the MnSOD is located in the chloroplast and is the dominant SOD in the cell [134]. Then, it is possible that the total CA activity in *T. pseudonana* relies on Mn^+2^ as well as Zn^2+^, Co^2+^, and Cd^2+^. Sunda and Huntsman [132] showed that the range of free Mn^2+^ concentration in which there is a cellular regulation of Mn^2+^ is related to the Mn^2+^ concentration found in the natural habitats where diatoms live.

Response to Mn^2+^ availability at the genetic level has been studied in several organisms. In the bacterium *Agrobacterium tumefaciens*, 55 genes were differentially expressed in manganese-limited cells, and the cells showed a reduction in biofilm formation [137]. Furthermore, the Mn-sensing transcriptional regulator MtsR controls the expression of genes related to Mn^2+^ uptake as well as genes controlling *Streptococcus pyrogenes* virulence [138]. In cyanobacteria, a Mn-sensing signaling system, Hik27-Rre16, regulates Mn^2+^ homeostasis by controlling the expression of a Mn^2+^-specific ABC transporter in response to Mn^2+^ availability [139,140]. In *C. reinhardtii* grown at limited Mn^2+^ concentrations, there is a strong down-regulation of genes involved in photosystem II function and up-regulation of the MnSOD gene; cells also showed defective photosynthesis and a loss of MnSOD activity [141]. It is possible that Mn has an effect on the expression of ι-CA, and this is currently under investigation in Gontero’s group.

## 7. Conclusions

The first occurrence of CA (an α-CA) in plants was confirmed in 1939. The discovery of β-CAs in plants in 1990 continued with the finding of multiple α- and β-CAs in *C. reinhardtii* and *A. thaliana* [18]. Subsequently, there has been a resurgence of interest in CAs from plants and algae over the past decade. Recent work on marine diatoms has uncovered further distinct classes of CA, some of which make use of metal cations other than zinc at the active site including ι-CA, which is widespread among marine phytoplankton, bacteria, and archaea. CA’s diversity, ubiquitous distribution, and multiple forms and location within an organism are testament to the fundamental importance of this enzyme to life on Earth. CA not only has consequences for global productivity but also for the biogeochemistry of trace metals in the ocean. The availability of bioinformatic tools to analyze the ever-growing number of sequenced genomes from algal species will help the global diversity and distribution of CAs to be investigated. Molecular approaches such as overexpression, gene silencing by CRISP/Cas9, and structural studies will shed light on the numerous metabolic roles of CAs in photosynthetic organisms.

## Figures and Tables

**Figure 1 ijms-21-02922-f001:**
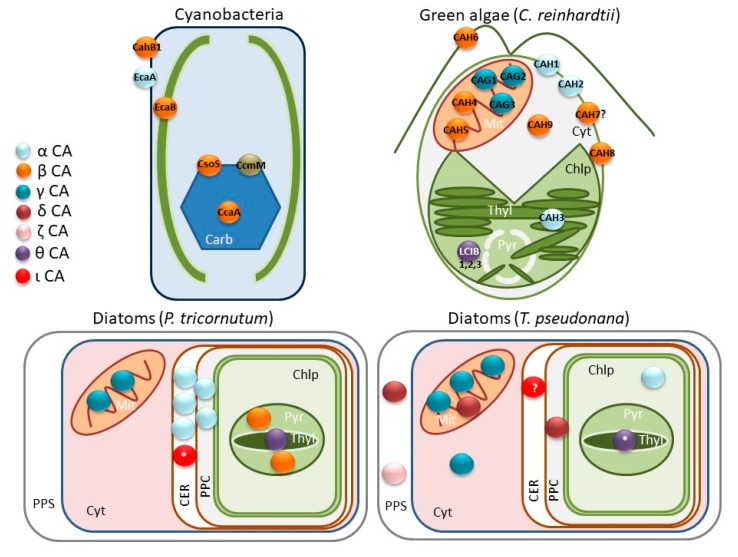
Distribution of the predicted carbonic anhydrases (CA) classes in some microalgae. Different CA classes are shown in colors as specified in the legend. In cyanobacteria and green algae, CAs appear also with their common names found in the literature. CAs whose subcellular localization is not fully demonstrated are shown with question marks (?); similarly, those that are predicted only by sequence homology appear with an asterisk (*). Carb: Carboxysome, Cyt: Cytoplasm, Mit: Mitochondrion, Chlp: Chloroplast (stroma), Thyl: Thylakoids (lumen), Pyr: Pyrenoid, PPS: Periplasmic space, CER: Chloroplast endoplasmic reticulum, PPC: Periplastidial compartment.

**Figure 2 ijms-21-02922-f002:**
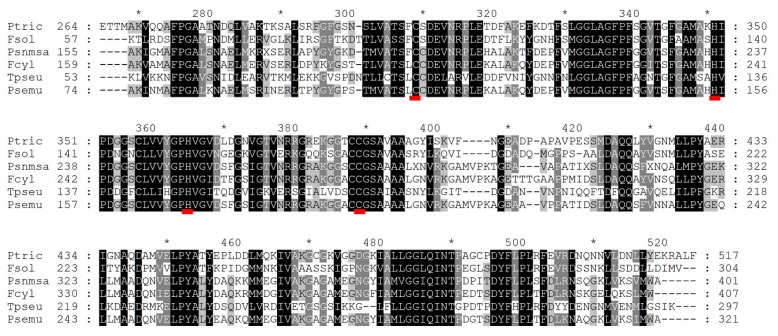
Alignment of the θ-CA from different diatom species. Only partial sequences are shown. Black, dark gray, and light gray indicate 80% or above, 70%, and 60% of amino acid identity, respectively. Species and protein IDs are the following: Ptric, *Phaeodactylum tricornutum* (protein ID: XP_002177507.1 – NCBI; 517 aa); Fsol, *Fistulifera solaris* (protein ID: GAX24004.1 – NCBI; 304 aa); Psnma, *Pseudo-nitzchia multistriata* (protein ID: VEU35824.1 – NCBI; 401); Fcyl, *Fragilariopsis cylindrus* (protein ID: OEU22620.1 – NCBI; 407 aa); Tpseu, *Thalassiosira pseudonana* (protein ID: XP_002297283.1 – NCBI; 297 aa); Psemu, *Pseudo-nitzchia multiseries* (protein ID: 239261 – JGI; 321 aa). The amino acids underlined in red are possibly involved in the active site, as shown by Kikutani et al. [11] Alignments were performed with ClustalW, using MEGAX software, and the figure was processed with GeneDoc.

**Table 1 ijms-21-02922-t001:** The different carbonic anhydrases (CA) classes, their metal cofactors, and distribution.

CA Class	Metal Cofactor	Organism(s)	References
**α-CA**	Zn^2+^	Mammals, plants, algae, prokaryotes	[18,19]
**β-CA**	Zn^2+^	Plants, algae, bacteria	[18]
**γ-CA**	Zn^2+^, Fe^2+^, Co^2+^	Prokaryotes, plants, fungi, algae	[20]
**δ-CA**	Zn^2+^, Co^2+^	Marine phytoplankton	[21,22]
**ζ-CA**	Cd^2+^, Zn^2+^	Diatoms	[23]
**η-CA**	Zn^2+^	*Plasmodium* sp	[16]
**θ-CA**	Zn^2+^	Diatoms, green algae	[11,24]
**ι-CA**	Mn^2+^	Marine phytoplankton	[1,12]

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
