# Peer review of "Insights on the Functions and Ecophysiological Relevance of the Diverse Carbonic Anhydrases in Microalgae"

_ijms, 2020, doi:10.3390/ijms21082922_

Round 1

Reviewer 1 Report

The review by Jensen et al. is of undoubted interest for the scientific community. The ms is well written and structured. As unique minor suggestions, I recommend to put in italic all nouns of species reported in the text and to check the correspondence between numbers and references as someone is not corresponding e.g. 15.

Author Response

We are very thankful of reviewer 1 for their positive feed back regarding our manuscript. We have considered their suggestions and made the appropriate modifications in our manuscript. Following, we show the reviewer’s comments in black and our answers in red color.

Reviewer #1 comments:

The review by Jensen et al. is of undoubted interest for the scientific community. The ms is well written and structured. As unique minor suggestions, I recommend to put in italic all nouns of species reported in the text and to check the correspondence between numbers and references as someone is not corresponding e.g. 15.

  • Line 54: the reference mistake was fixed. Now this is reference number 16 (Del Prete et al 2014. Discovery of a new family of carbonic anhydrases in the malaria pathogen Plasmodium falciparum - The η-carbonic anhydrases). We checked the correspondence of all our references as well.
  • We checked and corrected the writing of the scientific name of all species mentioned throughout the text (i.e. corrected to italic).

Reviewer 2 Report

This is a very interesting, complete and well written review. Some minor comments below:

- lines 48-49: references 8-13, should be more general, not only focused on ...
- line 73: "occurs" should be replaced by "occur"
- line 103: fix typo in "HCO3-uptake"
- line 120: remove ";"
- line 127: replace "distributed" by "distributed in photosynthetic organisms"
- line 129: more general references on carbonic anhydrase are required
- line 165: the sentence "Three of the eight known CA classes were first discovered in diatoms [12,13,15,88]." was already used lines 55-57.
- lines 175 and 178: "T. weissflogii" should be in italic font
- lines 183 and 185: "P. tricornutum" should be in italic font
- lines 187, 286, 287, 288: "C. reinhardtii" should be in italic font
- lines 191-192: "T. pseudonana, T. oceanica, Fragilariopsis cylindrus" and "Fistulifera solaris" should be in italic font
- line 210: Why Figure 2 doesn't show the complete N-term sequences? If the CA is just a domain in a more complex structure, this should be specifically mentioned, otherwise full sequences should be shown. Metal-coordinating residues should also be highlighted.
- line 255: replace "2" by "PtCA2"

As a general comment, I would suggest to also include citations for more general works in the field of CAs. Several books and book chapters were published in the recent years, and many review articles as well.

I recommend the publication of this manuscript in IJMS with minor revision, taking into account the comments mentioned above.

Author Response

We are very thankful of reviewer 2 for their positive feed back regarding our manuscript. We have considered all their suggestions and we have made the appropriate modifications to our manuscript. Following, we show the reviewer’s comments in black and our answers in red color. We hope these modifications will improve the quality of our work.

Reviewer #2 comments (answers in red):

This is a very interesting, complete and well written review. Some minor comments below:

- lines 48-49: references 8-13, should be more general, not only focused on ... R: We added more general references (Nocentini 2019 and Supuran 2018).

- line 73: "occurs" should be replaced by "occur" R: corrected.

- line 103: fix typo in "HCO3-uptake" R: (now line 160) corrected.

- line 120: remove ";" R: (now line 122) corrected.

- line 127: replace "distributed" by "distributed in photosynthetic organisms" R: (now line 129) the phrase was modified.

- line 129: more general references on carbonic anhydrase are required R: (now line 131) Two more references were added to include more general work (Chegwidden 2000 and Foster 2000).

- line 165: the sentence "Three of the eight known CA classes were first discovered in diatoms [12,13,15,88]." was already used lines 55-57. R: We considered to change the sentence in lines 55-56 and maintain that in line 165.

- lines 175 and 178: "T. weissflogii" should be in italic font R: (now lines 178 and 182) corrected.

- lines 183 and 185: "P. tricornutum" should be in italic font R: (now lines 186 and 188) corrected.

- lines 187, 286, 287, 288: "C. reinhardtii" should be in italic font R: (now lines 190, 276, 292, 293) corrected.

- lines 191-192: "T. pseudonana, T. oceanica, Fragilariopsis cylindrus" and "Fistulifera solaris" should be in italic font. R: (now lines 194 and 195) corrected.

- line 210: Why Figure 2 doesn't show the complete N-term sequences? If the CA is just a domain in a more complex structure, this should be specifically mentioned, otherwise full sequences should be shown. Metal-coordinating residues should also be highlighted. R: We did not consider adding the complete sequences since the N-terminus does not share high sequence it does not add any more information. In addition, we think it is better not to give the full-length sequences as the Figure would be too large. However, we agree with the reviewer’s suggestion regarding the necessity of highlighting the amino acids involved in the metal coordination (i.e. active site), so we did the appropriate modifications to the figure and we indicate these amino acids with a red mark. In addition, we added in the legend the total number of amino acids residues in the full-length sequence as well as the accession number of each sequence used in the alignment.

- line 255: replace "2" by "PtCA2" R: (now line 261) corrected.